# Absence of *Streptococcus pneumoniae* Capsule Increases Bacterial Binding, Persistence, and Inflammation in Corneal Infection

**DOI:** 10.3390/microorganisms10040710

**Published:** 2022-03-25

**Authors:** Mary A. Carr, Dennis Marcelo, K. Michael Lovell, Angela H. Benton, Nathan A. Tullos, Erin W. Norcross, Brandon Myers, Marcus K. Robbins, Hayley Craddieth, Mary E. Marquart

**Affiliations:** 1Department of Microbiology and Immunology, University of Mississippi Medical Center, Jackson, MS 39216, USA; mcarr3@umc.edu (M.A.C.); klovell@umc.edu (K.M.L.); mkrobbins@umc.edu (M.K.R.); 2Colorado Springs Interventional Pain Management, Colorado Springs, CO 80909, USA; dennis.r.marcelo@gmail.com; 3Lake Eerie College of Osteopathic Medicine, Bradenton, FL 34211, USA; abenton@lecom.edu; 4Department of Biological Sciences, Mississippi College, Clinton, MS 39056, USA; natullos@mc.edu (N.A.T.); norcross@mc.edu (E.W.N.); 5Department of Business, George Mason University, Fairfax, VA 22030, USA; brandonmyers100@gmail.com; 6Department of Biology, University of Southern Mississippi, Hattiesburg, MS 39401, USA; hayleycraddieth@gmail.com

**Keywords:** adhesion, capsule, inflammation, pneumococcus, ocular, *Streptococcus pneumoniae*

## Abstract

The role of the pneumococcal polysaccharide capsule is largely unclear for *Streptococcus pneumoniae* keratitis, an ocular inflammatory disease that develops as a result of bacterial infection of the cornea. In this study, capsule-deficient strains were compared to isogenic parent strains in their ability to adhere to human corneal epithelial cells. One isogenic pair was further used in topical ocular infection of mice to assess the contribution of the capsule to keratitis. The results showed that non-encapsulated pneumococci were significantly more adherent to cells, persisted in significantly higher numbers on mouse corneas in vivo, and caused significant increases in murine ocular IL9, IL10, IL12-p70, MIG, and MIP-1-gamma compared to encapsulated *S. pneumoniae*. These findings indicate that the bacterial capsule impedes virulence and the absence of capsule impacts inflammation following corneal infection.

## 1. Introduction

The outer polysaccharide capsule of *Streptococcus pneumoniae* (pneumococcus) is known to be important in the pathogenesis of pneumonia, otitis media, and septicemia. One of the crucial functions of the capsule is protection from phagocytosis, making it clear why non-encapsulated strains were widely considered to be avirulent until recently. In 1982, a report of an outbreak of conjunctivitis in New York identified “nontypable” *S. pneumoniae* as the putative cause [1], establishing non-encapsulated *S. pneumoniae* as a cause of bacterial conjunctivitis [2,3,4,5,6]. Although Weiser and colleagues noted in the 1990s that transparent (lacking or significantly reduced capsule) variants of pneumococci were better able than opaque (capsular) variants to colonize the nasopharynxes of rats [7], only recently have non-encapsulated pneumococci been highlighted as serving a potential role in virulence. In addition to the colonization of the nasopharynx [8,9], non-encapsulated strains can cause otitis media in children [8] and are postulated to either promote virulence of encapsulated strains or cause disease directly [10]. A recent study from Japan determined that 1.6% of pneumococcal isolates from non-invasive infections were non-encapsulated and 88.7% of these isolates possessed multidrug resistance [11]. A range of 4–19% has been estimated as the incidence of non-encapsulated carriage strains [12,13]. Invasive infections caused by non-encapsulated pneumococci are rare but concerning due to the overall trend of an increase in infections caused by strains that are not covered by the currently available vaccines [12,14,15]. One of the confirmed sources of ocular non-encapsulated *S. pneumoniae* is the nasopharynx; genotypic matches between nasopharyngeal carriage isolates and conjunctival isolates were reported in 87% of children positive for conjunctival *S. pneumoniae* [16].

Our first study of *S. pneumoniae* sought to examine the role of capsule in bacterial keratitis (infection of the cornea), as little was known regarding pneumococcal pathogenesis in the eye despite this organism being one of the more common causes of bacterial keratitis [17,18,19]. We had determined that a non-encapsulated strain was as virulent as the parental type 2 strain in a model of rabbit keratitis [20]. The limitation of the rabbit model of keratitis, however, is that bacteria must be injected directly into the cornea because rabbit eyes are resistant to topical infection with *S. pneumoniae*.

As the act of injecting the bacteria directly into the eye does not accurately replicate natural infection of *S. pneumoniae* we instead chose to use the mouse model of keratitis for this study. The mouse model of keratitis only requires scratching of the cornea prior to pipetting the bacteria onto the surface of the eyeball.

Keratitis, which typically develops following trauma to the eye, can lead to permanent scarring of the cornea [21]. The potential for long-term effects of keratitis and the ability of non-encapsulated bacteria to contribute to instances of this disease, make this an important research question. Additionally, non-encapsulated strains of *S. pneumoniae* have been shown to have a higher occurrence of multidrug resistance genes than encapsulated strains [22]. Possibly compounding this issue is the selective pressure provided by the pneumococcal conjugate vaccine. While this vaccine is effective at preventing pneumococcal disease, there are concerns that because it is specifically targeted against the polysaccharide capsule there will be an increase in infections caused by non-encapsulated strains [12,14,15]. Although the conjugate vaccine does not protect against conjunctivitis, we cannot rule out the possibility that the positive selection caused by the vaccine will have negative impacts on the rates and or severity of eye disease caused by the bacteria. 

Hammerschmidt et al. showed that encapsulated bacteria, in close proximity to host cells, will modulate their quantities of capsule. The group also showed that these same bacterial cells will revert to normal levels of capsule following entry or colonization [23]. This suggests that although the capsule is thought to be important to pneumococcal disease, at least in some circumstances a lack or reduction in capsule provides the bacteria with an advantage. We posit that non-encapsulated pneumococci are increased in their ability to adhere to the corneal surface. *S. pneumoniae* possesses over 80 surface proteins, of which a large portion are putative or confirmed adhesins of host cells [24]. Reduced or absent capsules, therefore, would likely enhance the ability of surface-exposed adhesins to bind to their host receptors.

Since pneumococcal keratitis is characterized by corneal opacity resulting from neutrophil infiltration [25], the increased adherence and persistence of non-encapsulated bacteria should result in an increase in inflammatory markers such as pro-inflammatory cytokines in the eye. Toll-like and Nod-like receptors have been shown to play roles in the host response to pneumococcal keratitis [25,26]. IL1-alpha, IL1-beta, and IFN-gamma are increased in human donor corneas of patients with *S. pneumoniae* keratitis [27]. However, a gap in knowledge of other inflammatory mediators such as pro- and anti-inflammatory cytokines during the progression of this disease exists. Therefore, the aim of the study described herein was to determine whether non-encapsulated pneumococci are more adherent to the cornea and induce more production of inflammatory markers compared to encapsulated *S. pneumoniae*. 

## 2. Materials and Methods

### 2.1. Bacterial Strains and Growth Conditions

Pneumococcal strains and their corresponding serotypes are listed in Table 1. A previous study identified K1544 as type 38 [28]; however, further analyses determined that K1544 is type 15B/C. Typing was achieved using a multiplex PCR method [29]. Strains were kindly provided by Larry S. McDaniel (University of Mississippi Medical Center, Jackson, MS, USA) and Regis P. Kowalski (Charles T. Campbell Ocular Microbiology Laboratory, University of Pittsburgh, Pittsburgh, PA, USA). Each strain was maintained as a frozen glycerol stock and was routinely isolated on blood agar with incubation at 37 °C and 5% CO_2_. Isolated colonies were cultured statically for approximately 16 h in Todd Hewitt broth containing 0.5% yeast extract (THY) at 37 °C and 5% CO_2_. These stationary phase cultures were then diluted 100-fold in fresh THY and incubated until reaching mid-logarithmic phase at an A_600_ determined by prior growth curves to correspond to approximately 10^8^ colony-forming units (CFU) per milliliter.

### 2.2. Host Cell Adhesion Assays

Immortalized human corneal epithelial cells (HCECs) were a kind gift from Haydee Bazan (Louisiana State University Eye Center, New Orleans, LA, USA) and were cultured as previously described [30]. Adhesion of each *S. pneumoniae* strain was quantitated using the method of Vanier et al. [31], except that logarithmic phase bacteria were used and were incubated with the HCECs for 2 h. HCECs from individual wells of assay plates were counted with a hemacytometer for each experiment to determine the quantity of bacteria yielding a target multiplicity of infection (MOI) of 1. Bacterial inocula were routinely serially diluted and plated on blood agar to verify quantity. Three technical replicates were performed for each experiment, serial dilutions of each replicate were plated in triplicate on blood agar, and experiments were repeated to ensure reproducibility. Serial dilutions were always vortexed to dissociate bacteria that clump because non-encapsulated strains are highly self-adherent. A 2-tailed *t*-test was performed to compare D39 with D39Δ*cps*, or K1544 with K1544Δ*cps*, in terms of the mean log_10_ CFU/mL bound to HCECs.

### 2.3. Transmission Electron Microscopy

Each bacterial strain was cultured to logarithmic phase in THY and collected by centrifugation. Bacterial pellets were fixed in 0.1 M phosphate buffer pH 7.0 containing 2% glutaraldehyde and stained with Alcian blue. Transmission electron microscopy (TEM) was performed as a service by Glenn Hoskins (Department of Pathology, University of Mississippi Medical Center) with a Zeiss TEM LEO 912. Capsule thickness was measured for each strain by selecting a single coccus or diplococcus in a field that appeared representative of all cells for a particular strain, then by taking the mean of a minimum of 10 measurements from the cell wall to the outer edge of the capsule.

### 2.4. Mouse Corneal Infection

Mouse experiments were performed according to the guidelines of the ARVO Statement for the Use of Animals in Ophthalmic and Vision Research and complied with the Public Health Service policy of the United States. Animal protocol number 1093E was reviewed and approved by the Institutional Animal Care and Use Committee of the University of Mississippi Medical Center.

Six-week-old male and female A/J mice (The Jackson Laboratory, Bar Harbor, ME, USA) were anesthetized with a mixture of ketamine and xylazine, then proparacaine was applied to one eye of each mouse. The cornea of that eye was then scratched to the superficial stroma in a cross-hatch pattern with a sterile 30-gauge needle, and 0.01 mL of approximately 10^8^ CFU logarithmic phase K1544 or K1544Δ*cps* was applied by pipette. Each mouse was monitored for 10 min to ensure the inoculum did not drop from the cornea. Bacterial inocula were routinely serially diluted and plated on blood agar to verify quantity.

Two types of in vivo experiments were performed. The first consisted of periodic clinical examination of the eyes over 10 days to assess inflammation. A 4-point scoring system was used as previously described [32], and the examiner was masked to the identity of the group designations during examination to reduce bias. Mice were euthanized after 10 days, and their whole eyes were collected for homogenization. Homogenized eyes were serially diluted and plated on blood agar to determine bacterial loads and to quantitate cytokines. The second type of experiment was an acute in vivo bacterial load determination at 4 time points with accompanying cytokine quantitation at 2 time points. At 4, 8, 12, and 24 h after infection, mice were euthanized and eyes were collected and homogenized for CFU quantitation. Three eyes each from 8 and 24 h were selected for cytokine quantitation. Uninfected scratched cornea control eyes were also collected. Each experiment was performed on more than one occasion with independently prepared bacterial inocula. 

A 2-tailed *t*-test was performed to compare K1544 with K1544Δ*cps* in terms of bacterial load recovered or clinical score for each individual time point; scratch control corneas had zero clinical scores, no culturable bacteria, and were not included in the analysis. A separate analysis was performed to examine potential sex differences, in which a one-way analysis of variance (ANOVA) was used to detect overall differences between 4 groups (female/K1544, female/K1544Δ*cps*, male/K1544, and male/K1544Δ*cps*).

### 2.5. Inflammatory Marker Quantitation

An antibody-based mouse inflammation array was used to quantitate 40 cytokines and inflammatory markers from the homogenized eyes of scratch control and infected mice (RayBiotech, Norcross, GA, USA). Three homogenized eyes each of scratch controls, K1544-infected, and K1544Δ*cps*-infected mice from 8 h, 24 h, and 10 days after infection were adjusted to equal total protein concentration. Each adjusted ocular homogenate (0.5 mg/mL) was applied to an individual array. The remainder of the assay was performed according to the manufacturer’s instructions (RayBiotech). 

Densitometry of each array was performed using the ImageJ plugin [33]. Densitometric reads were normalized using the RayBiotech analysis spreadsheet accompanying the array. Normalized reads were then subjected to one-way ANOVA to detect overall significant differences between the 3 groups (scratch control, K1544-infected, and K1544Δ*cps*-infected) for each marker, followed by pairwise comparisons using a 2-tailed *t*-test. 

## 3. Results

### 3.1. Capsule Thickness and Adhesion to Host Cells

TEM images of each strain are in Figure 1a. K1472 (type 15B/C) followed by K1451 (type 19A) had the greatest capsule thickness of all the strains (Figure 1b). Likewise, K1472 followed by K1451 had the greatest quantity of CFU bound to HCECs (Figure 1c). It should be noted that the capsule thickness quantitated for K1370 could be an anomaly since serotype 3 strains tend to produce copious capsular polysaccharides (as observed on blood agar) that exhibit non-uniform, globular structures in TEM [23].

### 3.2. Relative Adhesion of Non-Encapsulated Mutant Strains to Host Cells

TEM images of isogenic mutants deficient in their capsule loci confirm a lack of capsule production (Figure 2a). Of note, non-encapsulated pneumococci are highly self-adhesive accounting for the tightly packed K1544Δ*cps* image and the common laboratory observation of clumping and settling in liquid media.

Adhesion to HCECs was assessed for D39 compared to D39Δ*cps*, and K1544 compared to K1544Δ*cps*. Each non-encapsulated isogenic mutant had significantly increased CFU bound to HCECs (Figure 2b; *n* = 12 each for D39 and D39Δ*cps*; *n* = 6 each for K1544 and K1544Δ*cps*; *p* < 0.001 for each comparison of parent strain and isogenic mutant).

### 3.3. Mouse Corneal Infection

Clinical keratitis strain K1544 and its isogenic mutant, K1544Δ*cps*, were selected for in vivo experiments. Keratitis was generally mild in mice as shown by low clinical scores and eye photographs in Figure 3. However, a significant difference in clinical scores was calculated for 1 day after infection (Figure 3a; *n* = 22 for K1544, *n* = 23 for K1544Δ*cps*; *p* = 0.033). Analysis of scores in the context of sex yielded an overall *p*-value of 0.073, with the trend indicating males, but not females, to be more susceptible to K1544Δ*cps* over K1544. Since all eyes were sterile by the experimental endpoint of 10 days regardless of strain, we performed acute in vivo growth determinations to assess whether the clinical score difference at day 1 could be due to bacterial load differences.

K1544 loads in mouse eyes were significantly more reduced than those of K1544Δ*cps* 4 (*p* = 0.014) and 8 h (*p* = 0.024) after infection (Figure 4; *n* = 7, 8, 8, and 7 for K1544 at 4, 8, 12, and 24 h, respectively; *n* = 8 per time point for K1544Δ*cps*). K1544 loads decreased by approximately 2.5 and 3.5 log_10_ CFU at 4 and 8 h, respectively, whereas K1544Δ*cps* loads decreased by about 2 and 2.5 log_10_ CFU at those time points. Analysis of bacterial recovery separated by sex resulted in *p* < 0.05, and pairwise testing indicated a significant difference between K1544 and K1544Δ*cps* in females but not males at 4 h (*p* = 0.046; *n* = 4 per sex per strain except *n* = 3 males for K1544) and males but not females at 8 h (0.007; *n* = 4 per sex per strain).

### 3.4. Inflammatory Markers

Three eye homogenates, irrespective of sex, were selected from each of 3 time points for analysis. Selection was prioritized on the proximity of a cornea’s bacterial load (8 and 24 h time points) or clinical score (10 day time point) to its respective group mean. Scratched but uninfected eyes were included in the analysis to control for the effects of the corneal scratching. Forty inflammation-associated markers were quantitated by antibody arrays (listed in Table 2).

There were no significant differences between K1544- and K1544Δ*cps*-infected eyes in the relative quantities of each marker 8 h after infection (data not shown). However, K1544Δ*cps*-infected eyes had significantly increased quantities of IL9, IL10, IL12-p70, MIG, and MIP-1-gamma compared to K1544-infected and scratch control eyes 24 h after infection (Figure 5; *p* = 0.037, 0.007, 0.029, 0.037, and 0.036, respectively). There were no significant differences detected 10 days after infection (data not shown).

## 4. Discussion

The findings from this study suggest that the pneumococcal capsule inhibits binding to the cornea in the early stages of keratitis. Although the long-term disease produced by K1544 was mild in A/J mice relative to a different clinical strain from our previous work [32], we detected a significant increase in mean clinical score and five inflammation-associated markers for K1544Δ*cps*-infected eyes compared to wild type 24 h after infection. This difference in corneal opacity and cytokine production is likely due to the enhanced ability of K1544Δ*cps* to bind to the corneal epithelium and persist in mouse eyes during the early hours after infection. 

We chose to employ an antibody array that tests multiple pro- and anti-inflammatory cytokines and other markers to test for those cytokines that have been associated with pneumococcal keratitis [27] as well as screen for previously untested markers. To date, Toll-like and Nod-like receptors have been implicated in the neutrophil response to pneumococcal keratitis [25,26], but the molecules involved in the response are under-characterized. Five of the forty markers tested were significantly increased in K1544Δ*cps*-infected eyes 24 h after infection: IL9, IL10, IL12-p70, MIG, and MIP-1-gamma. 

IL9 can be produced by several types of cells but has been associated mainly as being produced by T cells, especially Th2 and Th9 cells. This cytokine is known to be produced following IL33 signaling and has been implicated in disruption of the ocular surface barrier in allergic conjunctivitis [34]. The role of IL9 in bacterial keratitis is largely unknown. 

IL10 is an anti-inflammatory cytokine produced by many different types of cells of both the innate and adaptive immune systems. IL10 plays a role in protecting ocular tissue during the later (healing) stages of *Pseudomonas* keratitis, but increased IL10 is also associated with defective clearance of infectious microbes [35,36,37]. A study examining the response to *Pseudomonas* corneal infection reported a significant increase in IL10 24 h after infection in whole eyes compared to corneas and emphasized the need to consider whole ocular responses, not only corneal responses [38]. This prior finding highlights the potential need to examine the whole globe versus corneal responses in specific disease models and is an avenue to pursue in our ongoing research on *S. pneumoniae* keratitis. The role of IL10 in the eye has not been determined for *S. pneumoniae*, but for pneumococcal lung infection, this cytokine causes reduced inflammation concomitant with increased bacterial burden [39].

*S. pneumoniae* induces dendritic cells to produce the heterodimeric cytokine IL12-p70 [40]. IL12-p70 is synonymous with IL12 when referring to the bioactive cytokine, whereas IL12-p40/p70 refers to the p40 subunit of IL12, which is related to IL23. IL12 is important for controlling bacterial burden during *Pseudomonas* keratitis [41].

MIG has been identified as stimulated during herpes simplex virus type 1 HSV-1 keratitis [42,43,44]. Interestingly, MIG is a T helper cytokine induced by IFN-gamma and is a somewhat surprising cytokine to be increased in *S. pneumoniae* keratitis. Very little is known regarding cell-mediated responses to *S. pneumoniae* in the eye. Outside of the eye, *S. pneumoniae* has been shown to elicit a strong MIG response in murine pneumonia [45]. This response varies depending upon the capsule type or strain of *S. pneumoniae* with a serotype 3 strain eliciting high MIG increases and a serotype 8 strain eliciting a response but comparatively lower than that of the other strain [46]. 

The potential role of MIP-1-gamma (CCL9) in keratitis is unknown. MIP-1-gamma has, however, been reported to be increased for a prolonged period of time in murine brains following *S. pneumoniae* meningitis [47].

IL1-alpha, IL1-beta, and IFN-gamma were included in the arrays for this study, and no significant differences were detected for these cytokines. However, these cytokines were reported to have elevated expression in human donor corneas of patients with *S. pneumoniae* keratitis [27]. Additionally, IL1-beta, TNF, MCP-1 (CCL2), and MIP-1-alpha (CCL3) were determined to be upregulated in human bacterial corneal ulcers that included those caused by *S. pneumoniae* along with other bacterial species [48]. The discrepancy between our current findings and those of the previous human studies could be due to the length of time following initial infection to quantitation, which was 10.9 ± 14 days [27] or 7–20 days [48] for the human corneas, inherent differences between mice and humans in responses, and/or the fact that the human studies quantitated markers from corneas whereas we used whole globes. A common denominator between the disease in humans and mice is the prevalence of neutrophils in the cornea [27,32].

A question that remains to be answered is whether lack of, or down-regulation of, capsule is pertinent to the pathogenesis of pneumococcal keratitis in humans. The overwhelming majority of *S. pneumoniae* that cause conjunctivitis are non-encapsulated due to disruption of the capsule biosynthetic locus [5,8]. Non-encapsulated conjunctivitis strains are phylogenetically distinct from the other non-encapsulated pneumococci that possess unique sets of genes in place of the capsule locus [10,49] and have an integrative conjugative element with a gene encoding a surface protein and putative adhesin [49]. The studies reporting the capsule status in keratitis indicate that encapsulated strains are the etiologic agents [49,50,51,52,53,54]. Moreover, 13 keratitis strains received from the Charles T. Campbell Ocular Microbiology Laboratory (Pittsburgh, PA, USA) are all encapsulated. Given that the status of keratitis strains leans toward their being encapsulated, an assumption might be made that capsules would not impede corneal infection. However, Hammerschmidt et al. demonstrated by electron microscopy that pneumococcal capsules were either reduced or absent when in close contact with lung epithelial cells [23], suggesting that lack of capsule would be advantageous for the bacteria to adhere to the cells. A limitation of our current study is that the capsule thicknesses of our clinical strains, which appeared to correlate with HCEC binding, were measured for bacteria grown in vitro in the absence of human cells. Future work should include measuring capsule expression and/or production in the presence of host cells to determine whether encapsulated keratitis strains dampen capsule production in the vicinity of corneal cells to promote binding.

In conclusion, our hypothesis that non-encapsulated *S. pneumoniae* would be more adherent to corneal cells was supported by an increase in adherence in vitro and increased persistence in vivo. The decreased clearance of non-encapsulated bacteria from the mouse cornea was accompanied by higher clinical scores and increased production of five cytokines. Determination of the roles of these cytokines during keratitis, whether the capsule is downregulated in encapsulated strains while in close proximity to corneal cells, and which adhesin(s) are involved in host cell binding, will aid in the understanding of how pneumococci adapt to the host environment.

## Figures and Tables

**Figure 1 microorganisms-10-00710-f001:**
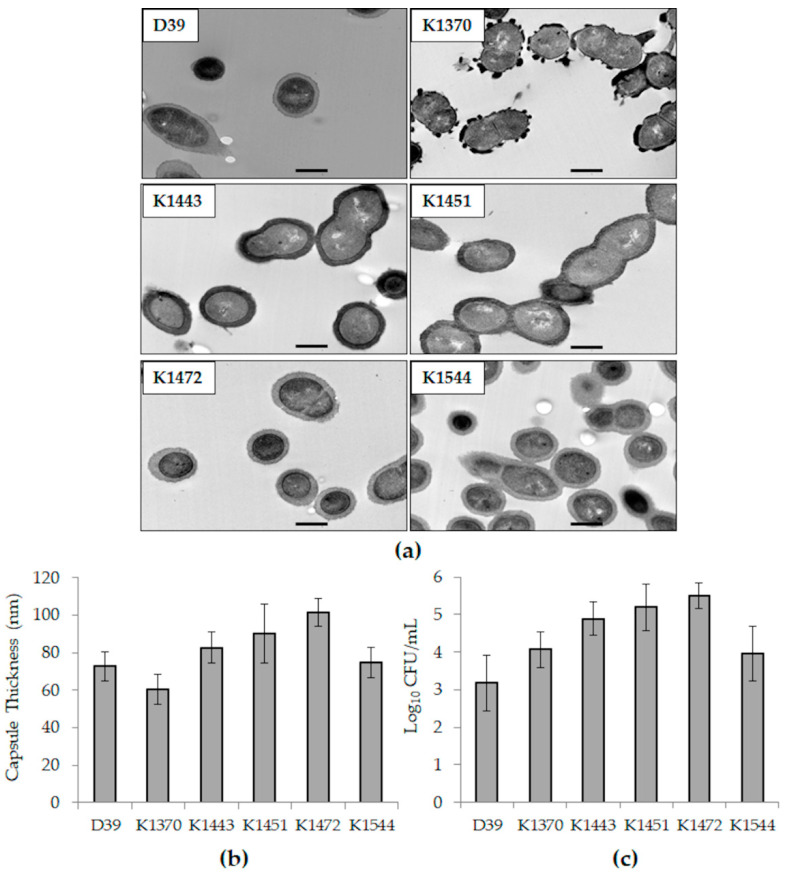
Capsules thickness and relative host cell adhesion of the strains included in this study. (**a**) TEM images of each strain. The bar in each panel represents a length of 500 nm; (**b**) average capsule thicknesses as measured from TEM images; (**c**) average log CFU/mL bound to HCECs. Error bars represent standard deviation.

**Figure 2 microorganisms-10-00710-f002:**
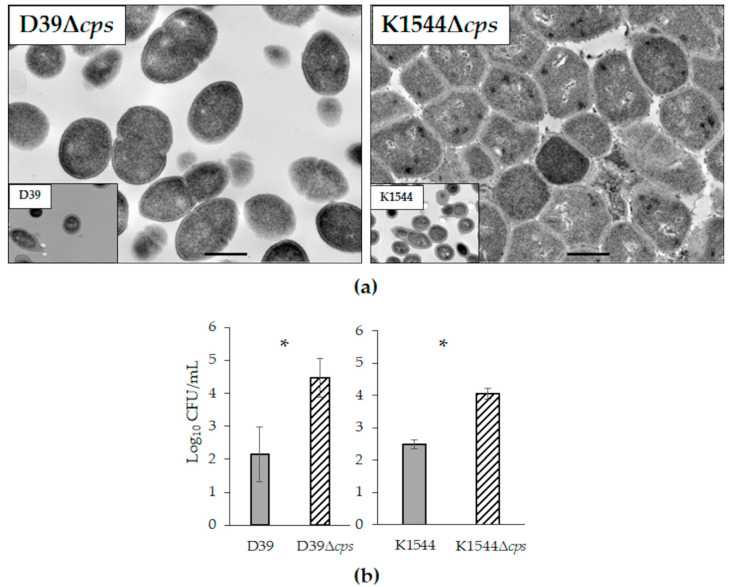
Isogenic non-encapsulated mutants of strains D39 and K1544. (**a**) TEM images of each non-encapsulated strain, with an inset of the parent strain for comparison. The bar in each panel represents a length of 500 nm; (**b**) average log CFU/mL bound to HCECs. Error bars represent standard deviation. Each asterisk indicates a significant difference between parent and mutant.

**Figure 3 microorganisms-10-00710-f003:**
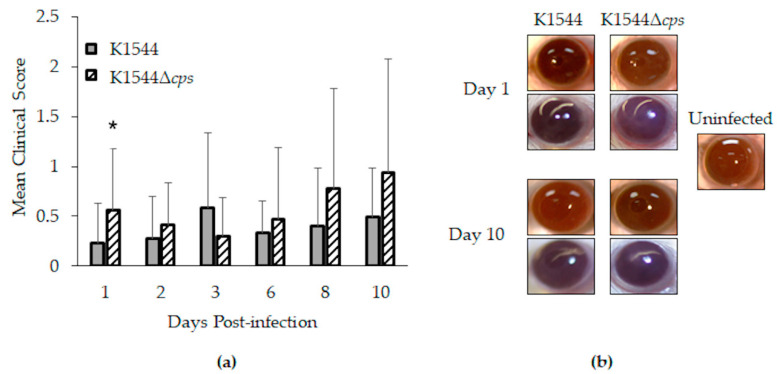
Keratitis in mice. (**a**) Mean clinical scores as determined by degree of corneal opacity on a scale of 0 to 4. Error bars represent standard deviation. Asterisk indicates a significant difference between parent and mutant; (**b**) representative photographs (2 for each strain at each time point shown). Corneal opacity was nearly undetectable for eyes infected with K1544, as demonstrated by clear visualization of pupils. For those infected with K1544Δ*cps*, opacity was detected as shown by slight obscurity of the pupils (bottom eye for Day 1 and bottom eye for Day 10).

**Figure 4 microorganisms-10-00710-f004:**
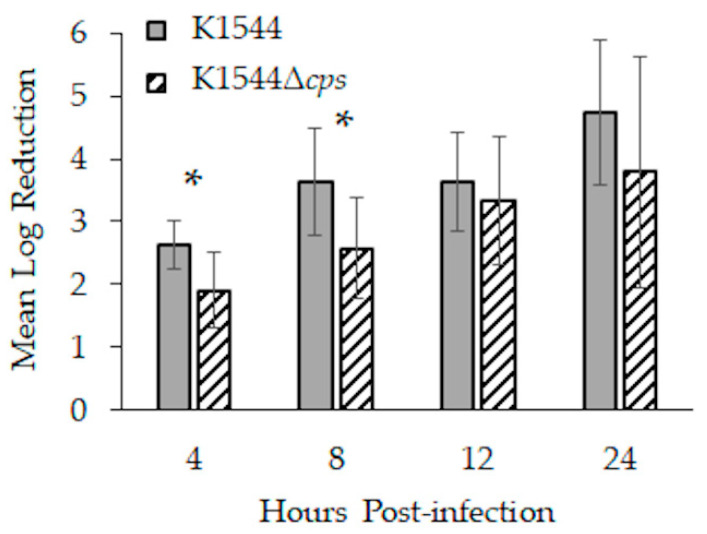
Bacterial load reductions during the first 24 h of infection in mice. Corneas were infected with approximately 8 log_10_ CFU, and each data bar indicates by how many log_10_ CFU that starting inoculum decreased. Error bars represent standard deviation. Asterisks indicate a significant difference between parent and mutant.

**Figure 5 microorganisms-10-00710-f005:**
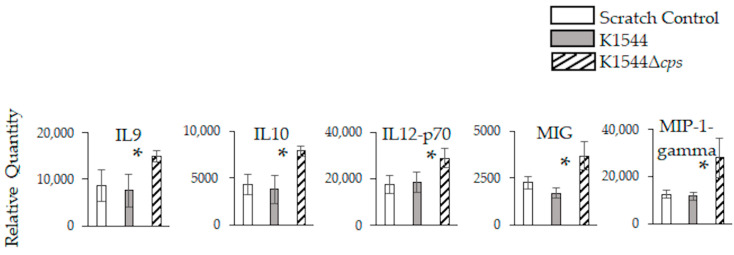
Inflammation-associated markers in mouse eyes 24 h after infection. Relative quantities are expressed as intensity values generated by ImageJ software. Error bars represent standard deviation. Asterisks indicate a significant difference between parent and mutant. Significant differences were detected only at 24 h.

**Table 1 microorganisms-10-00710-t001:** *S. pneumoniae* strains used in this study.

Strain	Capsule Type	Source or Reference
D39	2	L.S. McDaniel
D39Δ*cps*	−	L.S. McDaniel
K1370	3	Campbell Lab
K1443	19A	Campbell Lab
K1451	19A	Campbell Lab
K1472	15B/C	Campbell Lab
K1544	15B/C	Campbell Lab
K1544Δ*cps*	−	[28]

**Table 2 microorganisms-10-00710-t002:** Inflammation array proteins.

Name	Abbreviation	Alternate Name
B lymphocyte chemoattractant	BLC	C-X-C motif chemokine ligand 13; CXCL13
Cluster of differentiation 30 ligand	CD30L	Tumor necrosis factor superfamily 8; TNFSF8
Eotaxin	−	C-C motif chemokine ligand 11; CCL11
Eotaxin-2	−	C-C motif chemokine ligand 24; CCL24
Fas ligand	FasL	Tumor necrosis factor superfamily 6; TNFSF6
Fractalkine	−	C-X-3-C motif chemokine ligand 1; CX3CL1
Granulocyte-colony stimulating factor	G-CSF	−
Granulocyte-macrophage colony stimulating factor	GM-CSF	−
Interferon-gamma	IFN-gamma	−
Interleukin 1-alpha	IL1-alpha	−
Interleukin 1-beta	IL1-beta	−
Interleukin 2	IL2	−
Interleukin 3	IL3	−
Interleukin 4	IL4	−
Interleukin 6	IL6	−
Interleukin 9	IL9	−
Interleukin 10	IL10	−
Interleukin 12-p40/p70	IL12-p40/p70	−
Interleukin 12-p70	IL12-p70	−
Interleukin 13	IL13	−
Interleukin 17	IL17	−
Interferon-gamma-inducible T cell alpha chemoattractant	I-TAC	C-X-C motif chemokine ligand 11; CXCL11
Keratinocyte-derived cytokine	KC	C-X-C motif chemokine ligand 1; CXCL1
Leptin	−	−
Lipopolysaccharide-inducible C-X-C chemokine	LIX	−
Lymphotactin	−	−
Monocyte chemoattractant protein 1	MCP-1	C-C motif chemokine ligand 2; CCL2
Macrophage colony stimulating factor	M-CSF	−
Monokine induced by gamma	MIG	C-X-C motif chemokine ligand 9; CXCL9
Macrophage inflammatory protein-1-alpha	MIP-1-alpha	C-C motif chemokine ligand 3; CCL3
Macrophage inflammatory protein-1-gamma	MIP-1-gamma	C-C motif chemokine ligand 9; CCL9
Regulated upon activation, normal T cell expressed and secreted	RANTES	C-C motif chemokine ligand 5; CCL5
Stromal cell-derived factor 1	SDF-1	C-X-C motif chemokine ligand 12a; CXCL12a
T cell activation 3	TCA-3	C-C motif chemokine ligand 1; CCL1
Thymus-expressed chemokine	TECK	C-C motif chemokine ligand 25; CCL25
Tissue inhibitor of metalloproteinases-1	TIMP-1	−
Tissue inhibitor of metalloproteinases-2	TIMP-2	−
Tumor necrosis factor alpha	TNF-alpha	−
Soluble tumor necrosis factor receptor I	sTNF RI	−
Soluble tumor necrosis factor receptor II	sTNF RII	−

## Data Availability

Data are contained within the article.

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
