# Peer review of "Absence of Streptococcus pneumoniae Capsule Increases Bacterial Binding, Persistence, and Inflammation in Corneal Infection"

_microorganisms, 2022, doi:10.3390/microorganisms10040710_

Round 1
Reviewer 1 Report
Congratulations on your work!
It would be interesting to continue the studies in order to elucidate the problems related to the production of the capsule in the presence of the host cells.
Please consider whether "pneumococcus", as mentioned in parentheses (line 28), should not be capitalized (Pneumococcus).
Author Response
We thank the editors and reviewers for their efficient and helpful reviews of our research article entitled, “Absence of Streptococcus pneumoniae capsule increases bacterial binding, persistence, and inflammation in corneal infection.” Herein we detail our point-by-point responses to their reviews. Please note the line number indications in this response are based on the “Show Markup” version of the manuscript using the “Track Changes” tool. Sometimes the line numbers change during download on different operating systems, so we would recommend reading the marked up PDF version to avoid confusion.
Reviewer 1
- Congratulations on your work!
It would be interesting to continue the studies in order to elucidate the problems related to the production of the capsule in the presence of host cells.
Answer: Thank you very much for your positive feedback. We are excited to continue this work!
- Please consider whether “pneumococcus”, as mentioned in parentheses (line 28), should not be capitalized (Pneumococcus).
Answer: According to the World Health Organization and the Centers for Disease Control and Prevention, “pneumococcus” is not capitalized. We will be happy to change the term to capitalized if this reviewer wishes, but at this time we will keep the term lowercase.

Reviewer 2 Report
Carr et al
This is an interesting and generally well presented paper examines the role of capsule in attachment of pneumococcal strains to mouse cornea. The paper investigates an important infection area, but is rather imbalanced in its presentation, particularly its emphasis on cytokine profiling.
Specific points
Introduction
Line 41-43: ‘ Our first study of S. pneumoniae sought to examine the role of capsule in bacterial 41 keratitis (infection of the cornea), as little was known regarding pneumococcal pathogenesis in the eye despite this organism being one of the more common causes of bacterial keratitis.’
This sentence needs at least to be supported by a reference citation given the ocular keratitis subject of the paper.
Another subject not mentioned but relevant to this study’s focus, is how widespread non-encapsulated pneumococcal strains are in the environment? The capsule plays an important role in survival of the pneumococcus, so is this study truly relevant to pneumococcal eye infections?
Results
Line 193-194 Figure 3
This figure’s legend needs more explanation – what is the clinical score ie how was it measured in (a) and what are the images in (b) supposed to be showing?
Line 228-239 Figure 5
This overly figure needs improvement in its presentation and the data could be more effectively presented in numerical form in a table.
Discussion
There is too much discussion of the cytokine data and not enough of mechanism(s) which might underlie the apparently greater attachment to eye tissue of non-encapsulated pneumococcal strains.
Author Response
We thank the editors and reviewers for their efficient and helpful reviews of our research article entitled, “Absence of Streptococcus pneumoniae capsule increases bacterial binding, persistence, and inflammation in corneal infection.” Herein we detail our point-by-point responses to their reviews. Please note the line number indications in this response are based on the “Show Markup” version of the manuscript using the “Track Changes” tool. Sometimes the line numbers change during download on different operating systems, so we would recommend reading the marked up PDF version to avoid confusion.
Reviewer 2
- This is an interesting and generally well presented paper examines the role of capsule in attachment of pneumococcal strains to mouse cornea. The paper investigates an important infection area, but is rather imbalanced in its presentation, particularly on cytokine profiling.
Answer: We agree and thank this reviewer for the constructive suggestions to improve this manuscript. We have revised the presentation of the manuscript accordingly.
- Introduction
Line 41-43: ‘Our first study of S. pneumoniae sought to examine the role of capsule in bacterial keratitis (infection of the cornea), as little was known regarding pneumococcal pathogenesis in the eye despite this organism being one of the more common causes of bacterial keratitis.’
This sentence needs at least to be supported by a reference citation given the ocular keratitis subject of the paper.
Answer: We have incorporated relevant citations (line 53).
- Another subject not mentioned but relevant to this study’s focus, is how widespread nonencapsulated pneumococcal strains are in the environment? The capsule plays an important role in survival of the pneumococcus, so is this study truly relevant to pneumococcal eye infections?
Answer: Currently, nonencapsulated strains are still much lower in incidence compared to encapsulated strains, but their incidence is increasing. The concern over their increase is that they tend to be multidrug resistant and cannot be prevented by the currently available vaccines that are composed of capsular polysaccharides. Most of the non-encapsulated strains are isolated from the nasopharynx and are considered carriage isolates. These isolates were shown to be genotypic matches to the conjunctival isolates in 87% of children with ocular pneumococci. We have added additional information as to the incidence of non-encapsulated pneumococci in the environment (ie, the nasopharynx), as well as describe a study that showed ocular isolates to likely originate from the nasopharynx (lines 40-49).
- Results
Line 193-194 Figure 3
This figure’s legend needs more explanation – what is the clinical score ie how was it measured in (a) and what are the images in (b) supposed to be showing?
Answer: We have added the requested information (lines 218-223).
- Line 228-239 Figure 5
This overly figure needs improvement in its presentation and the data could be more effectively presented in numerical form in a table.
Answer: Thank you! We simplified the figure by omitting all panels that displayed non-significant differences. Our preference is to display the data in the current graph form but will change it to table form if our modifications/simplifications are not acceptable. The new Figure 5 is between lines 261 and 262.
- Discussion
There is too much discussion of the cytokine data and not enough of mechanism(s) which might underlie the apparently greater attachment to eye tissue of non-encapsulated pneumococcal strains.
Answer: The new paragraph added at the end of the Discussion (lines 358-365) recapitulates the proposed mechanism that is included in the new material at the end of the Introduction (lines 78-93). Although we don’t yet know the signaling mechanisms behind the increased attachment, we hypothesize that reduction of capsule promotes exposure of putative surface adhesins to their host cell receptors. We have kept the cytokine discussion because there is a paucity of understanding regarding the inflammation signals during pneumococcal keratitis. The special issue to which we submitted this manuscript is “Infection-induced chronic inflammation in different compartments of the eye,” so we believe it is important to include a discussion of the cytokines. We think that the additions and improvements we have made to both the Introduction and Discussion help balance the topics presented so that the manuscript is no longer disproportionate in the discussion of cytokines.

Reviewer 3 Report
Please see attached file.

Author Response
We thank the editors and reviewers for their efficient and helpful reviews of our research article entitled, “Absence of Streptococcus pneumoniae capsule increases bacterial binding, persistence, and inflammation in corneal infection.” Herein we detail our point-by-point responses to their reviews. Please note the line number indications in this response are based on the “Show Markup” version of the manuscript using the “Track Changes” tool. Sometimes the line numbers change during download on different operating systems, so we would recommend reading the marked up PDF version to avoid confusion.
Reviewer 3
- The paper "Absence of Streptococcus pneumoniae capsule increases bacterial binding, persistence, and inflammation in corneal infection" is interesting because it was commonly considered that the presence of the capsule is one of the virulence factors of S. pneumoniae. It is therefore important to show that S. pneumoniae without the capsule is a pathogen capable of causing infection.
In the reviewer's opinion, the paper was not prepared in a logical and orderly manner.
Answer: We have revised the manuscript according to the suggestions of this reviewer and the other reviewers. We formulated a clear aim and hypothesis (lines 78-93) and tied them to a new concluding paragraph (lines 358-365). We also offered more explanation of the intent behind the cytokine arrays (lines 286-291) and eliminated the over-discussion of sex as a potential variable. We believe that this manuscript is much improved and thank the reviewers for their comments.
- “The aim of the study described herein was to investigate the importance of S. pneumoniae polysaccharide capsule in a pneumococcal keratitis topical infection model using an isogenic capsule-deficient mutant”
The aim of the study was not precisely formulated. The expression "the importance of S. pneumoniae polysaccharide capsule in a pneumococcal keratitis" is too general. The assumptions of this work are not fully understood due to lack of basic information, e.g. justification of the purpose of performing specific tests to confirm the pathogenicity of S. pneumoniae strains. E.g. why the Authors performed determinations of indicated 40 parameters of immunological response? what was the aim of simultaneous determination of proinflammatory and anti-inflammatory cytokines? Performing too many tests without justification of their necessity and lack of detailed discussion of the test results makes the paper unclear.
Answer: We have clarified the aim of the study and used more specific language with a logical progression (Introduction lines 78-93). Regarding the cytokine arrays, we have included the reasoning for using these arrays rather than selecting individual cytokines to quantitate (Discussion lines 286-291). Specifically, very little is known regarding the host response to pneumococcal keratitis besides neutrophil infiltration being the main cause of corneal opacity. These cytokine arrays are ideal for casting a wide net if one does not know what specific cytokine to test. We chose the array described because it included common pro-inflammatory cytokines one might suspect to be involved in neutrophil recruitment (eg, IL1-beta, IL-6).
- Discussion
The first part of the discussion was focused on the comparison of infection in relation to sex, although the concept of the paper did not include this correlation.
Answer: This is a good point. We deleted the lengthy discussion of sex from the Discussion since it was not the main focus.
- The next part of the discussion concerns the role of parameters of the immune system in infection - this information should have been included in the Introduction.
Answer: We added information to the Introduction regarding immune system molecules (lines 83-90).
- No conclusion. As a result, the paper does not give an answer to the question whether the thesis put forward by the Authors was confirmed in the research.
Answer: We added a concluding paragraph to the Discussion (lines 358-365).
- In the reviewer's opinion, the paper should be rewritten and organized.
Answer: Thank you so much for your helpful comments. We hope that the revisions have improved the clarity and organization of our manuscript.

Round 2
Reviewer 2 Report
Although the authors have satisfactorily addressed some of the required revisions, there is still one outstanding. Figure 5 and its myriad tiny histograms is largely unchanged and reduces the impact of the paper's key message. Therefore conversion to a table is needed.
Reviewer 3 Report
The corrections of the text have greatly improved the quality of the manuscript. It may be published in its present form.